# Overexpression of Orange Gene (*OsOr*-R115H) Enhances Heat Tolerance and Defense-Related Gene Expression in Rice (*Oryza sativa* L.)

**DOI:** 10.3390/genes12121891

**Published:** 2021-11-26

**Authors:** Yu Jin Jung, Ji Yun Go, Hyo Ju Lee, Jung Soon Park, Jin Young Kim, Ye Ji Lee, Mi-Jeong Ahn, Me-Sun Kim, Yong-Gu Cho, Sang-Soo Kwak, Ho Soo Kim, Kwon Kyoo Kang

**Affiliations:** 1Division of Horticultural Biotechnology, Hankyong National University, Anseong 17579, Korea; yuyu1216@hknu.ac.kr (Y.J.J.); rhwldbs60@naver.com (J.Y.G.); ju950114@naver.com (H.J.L.); kgfood1@hanmail.net (J.S.P.); ajswl1202@naver.com (J.Y.K.); lyj7776@naver.com (Y.J.L.); 2Institute of Genetic Engineering, Hankyong National University, Anseong 17579, Korea; 3College of Pharmacy and Research Institute of Life Sciences, Gyeongsang National University, Jinju 52828, Korea; amj5812@gnu.ac.kr; 4Department of Crop Science, Chungbuk National University, Cheongju 28644, Korea; kimms0121@cbnu.ac.kr (M.-S.K.); ygcho@cbnu.ac.kr (Y.-G.C.); 5Plant Systems Engineering Research Center, Korea Research Institute of Bioscience and Biotechnology (KRIBB), Daejeon 34141, Korea; sskwak@kribb.re.kr (S.-S.K.); hskim@kribb.re.kr (H.S.K.)

**Keywords:** transgenic rice, heat tolerance, *OsOr-*R115H, gene expression

## Abstract

In plants, the orange (*Or*) gene plays roles in regulating carotenoid biosynthesis and responses to environmental stress. The present study investigated whether the expression of rice *Or* (*OsOr*) gene could enhance rice tolerance to heat stress conditions. The *OsOr* gene was cloned and constructed with *OsOr* or *OsOr*-R115H (leading to Arg to His substitution at position 115 on the OsOr protein), and transformed into rice plants. The chlorophyll contents and proline contents of transgenic lines were significantly higher than those of non-transgenic (NT) plants under heat stress conditions. However, we found that the levels of electrolyte leakage and malondialdehyde in transgenic lines were significantly reduced compared to NT plants under heat stress conditions. In addition, the levels of expression of four genes related to reactive oxygen species (ROS) scavenging enzymes (*OsAPX2*, *OsCATA*, *OsCATB*, *OsSOD-Cu/Zn*) and five genes (*OsLEA3*, *OsDREB2A*, *OsDREB1A*, *OsP5CS*, *SNAC1*) responded to abiotic stress was showed significantly higher in the transgenic lines than NT plants under heat stress conditions. Therefore, *OsOr-R115H* could be exploited as a promising strategy for developing new rice cultivars with improved heat stress tolerance.

## 1. Introduction

Global warming is expected to significantly reduce crop yields due to high temperatures that negatively affect plant growth and development. Thermal stress can eventually lead to cell damage and death in plants due to accumulation of excess reactive oxygen species (ROS), membrane lipid peroxidation and metabolic disturbance [1,2]. Under elevated temperature, plants have shown physiological symptoms such as scorching of leaves and stems, leaf abscission and senescence, shoot and root growth inhibition or fruit damage, which lead to a decreased productivity and quality of crops [3]. Until now, numerous heat stress defense genes have been reported to mainly encode signaling molecules such as mitogen activated protein kinase (MAPK) cascades, cellular protective enzymes such as peroxidase (POD), transcription factors such as heat shock transcription factors (HSFs), and cellular stress proteins such as heat shock proteins (HSPs) [4,5,6,7]. These genes have been reported to protect plant cells by enhancing resistance with maintaining ROS homeostasis in the cell or regulating various genes on the signal pathway under thermal stress [6,7]. It was reported recently that the function for carotenoids relates to the response of plants to environmental stress [4,5]. In addition, reactive oxygen species (ROS) generated under stress conditions oxidizes carotenoids and converts into various oxidation products including aldehydes, ketones, endoperoxides, and lactones [4]. This demonstrates that carotenoid accumulation and potential signaling in plant cells plays a positive role in heat stress response. The Orange (*Or*) gene encoding a cysteine-rich DnaJ protein has been reported to regulate carotenoid accumulation and abiotic stress resistance in various plant species [8,9,10,11,12,13,14,15,16]. The Or protein functions as a holdase chaperone, a post-transcriptional regulator of phytoene synthase (PSY), and a rate-limiting enzyme on the carotenoid biosynthetic pathway [12,17]. Moreover, Or protein has been reported to interact with oxygen-evolving enhancer protein 2-1 (PsbP) in the chloroplast and protect it from heat-induced denaturation [17,18]. An allele carrying a single nucleotide polymorphism (SNP), in which the 96th amino acid of the Or protein substituted arginine (Arg) to histidine (His), is associated with high-carotene accumulation in melon [9]. In addition, overexpression of AtOr protein (Q8VYD8; *AtOr*-R96H) leads to high level accumulation of carotenoids in *Arabidopsis* and tomatoes (*Solanum lycopersicum*) [13,19]. In the sequence comparison of the *Or* gene between the low-carotenoid and high-carotenoid lines in carrots (*Daucus carota*), SNP substituted from T to C has been identified, and it is reported that a codon TTG (leu) is changed to TCG (ser), resulting in high carotene accumulation [20]. In a recent experiment, we have found out that the site-directed mutagenesis of *IbOr* gene in sweet potato (*Ipomoea batatas* [L.] Lam), leading to Arg to His substitution at position 96 (based on the CmOr protein) in the IbOr protein (*IbOr*-R96H), showed significantly higher carotenoid contents and enhanced environmental stress tolerance [21,22]. In this study, to understand the physiological role of *OsOr* gene in rice, the carotenoid content and abiotic stress tolerance were investigated in overexpressing plants, *OsOr*-WT and *OsOr*-R115H. Our results show that in the overexpressing plants, *OsOr*-WT and *OsOr*-R115H, carotenoid content was not changed compared to NT plants, but heat stress tolerance was enhanced. In addition, these lines showed a high level of transcription of genes related to ROS scavenging enzymes under stress conditions. Interestingly, the *OsOr*-R115H transgenic line showed lower malondialdehyde (MDA), lower relative electrolyte leakage (REL), and higher proline content in leaves compared with NT plants. 

## 2. Materials and Methods

### 2.1. Growth Conditions and Stress Treatments 

Dongjin (*Oryza sativa* L.ssp. *japonica*) was used as a wild-type, cultivated in greenhouse facilities at Hankyong National University, Korea, as described by Jung et al. [23]. The expression levels of *OsOr* were investigated in roots, leaves, stems, panicles, and immature seeds at 7 days after pollination in Dongjin cultivar. The expression pattern of the *OsOr* in response to abiotic stresses including 48 °C heat, 200 mM NaCl, and 20% PEG6000 drought 100 μM GA_3_, 100 μM ABA, and 4 °C cold was monitored as previously reported by Park et al. [17]. Abiotic stress treatments were performed for 0 h, 3 h, 6 h, 12 h, and 24 h, and then samples were collected and stored at −80 °C. All experiments included three biological replicates.

### 2.2. Gene Cloning and Generation of Transgenic Rice

The full-length cDNA of *OsOr* (Os02g0651300) was amplified with specific primer sets for RT-PCR (Appendix A). The PCR product was cloned into pMD18-T vector (TAKARA, Seoul, Korea) and sequenced. Site-directed mutagenesis was performed to obtain point mutants by substituting H for the 115th amino acid R of the *OsOr* gene as previously described by Kim et al. [22]. Briefly, *OsOr*-R115H, which replaced the ORF (Open Reading Frame) (*OsOr*-WT) and major SNPs of the *OsOr* gene, was subcloned downstream of the binary vector pGWB5 expressed by the cauliflower mosaic virus (CaMV) 35S promoter, respectively. The constructed vector was transformed into *Agrobacterium tumefaciens* strain EHA105, which was introduced by infection with rice embryogenic callus, as previously described [24]. The transformed callus and plants were selected using 6 mg/L phosphinothricin and confirmed by PCR analysis, as previously reported [24]. Positive transgenic plants were first screened by PCR, and then further confirmed by TaqMan PCR to selection of single copy insertion of T-DNA and qPCR analysis to determine the expression of the transgene. In addition, the transgenic homo lines of T_3_ generation were used for the analytical experiment.

### 2.3. Phenotype Analysis

Pollen viability was evaluated as previously described [25] from transgenic plants with *OsOr* gene and wild-type plants at flowering time for pollen germination analysis. Nikon SMZ800N microscope (Nikon Metrology Inc., Seoul, Korea) was used to observe pollen germination. The amylose content of whole grain was measured using the AMYLOSE/AMYLOPECTIN kit (Megazyme Ltd., Bray, Ireland) according to the manual provided by the manufacturer. The morphology of starch grains was observed by scanning electron microscopy (SEM). Images were captured using a Quanta FEG 450 instrument (ZFE, Graz, Austria). *Xanthomonas oryzae* pv. *oryzae* (*Xoo*) inoculation and determination of bacterial populations were evaluated as previously described [26,27] for a wild-type plant and transgenic plants. In addition, 30 plants were randomly selected from each of the transgenic lines and NT plants, and the seed setting rate, pollen fertility, 1000-grain weight, and yield per plant were investigated.

### 2.4. Analysis of Carotenoid Contents

Carotenoids were extracted from rice leave tissues using 0.01% solution of butylatedhydroxytoluene in acetone and analyzed using an Agilent 1100 HPLC system (HewlettePackard, Palo Alto, CA, USA) according to the method described by Lim et al. [28].

### 2.5. Analysis of Heat Tolerance of Transgenic Rice

To induce heat stress, leaf discs were excised from the fourth leaves of 2-week-old transgenic and NT rice plants and treated to high temperature (48 °C) in the dark for 24 h.

To measure tissue levels of H_2_O_2_ and visualize the degree of damage caused by heat stress, leaf discs were immersed in 3,3’-diaminobenzidine (DAB) solution (1 mg/mL, pH 3.8) for 5 h at 25 °C under continuous light, as described previously [18]. To measure the extent of cellular damage or membrane disruption, ion leakage was quantified using an ion conductivity meter (MTD, Schwerzenbach, Switzerland). For quantitative analysis of total area of DAB staining region, images were processed using ImageJ.

### 2.6. Physiological Parameters Measurements of Transgenic Rice

To investigate whether the transgenic lines can be used as oxidative stress biomarkers, the relative electrolyte leakage (REL), proline, and malondialdehyde (MDA) content were detected based on the previously described methods. Briefly, 500 mg of rice leaves were pulverized in 2 mL of the chilled reagent (0.25% (w/v) thiobarbituric acid in 10% (w/v) trichloroacetic acid). The extracts were treated at 100 °C for 30 min, cooled at room temperature, and centrifuged at 12,000× *g* for 15 min. The absorbance of the supernatant was measured at 450, 532, and 600 nm. The MDA content was calculated according to the formula: 6.45 × (OD_532_ − OD_600_) − 0.559 × OD_450_ [29,30,31].

### 2.7. qRT-PCR Analysis

Total RNA was isolated from different tissues of rice plants using a RNeasy plant mini kit (Qiagen, Seoul, Korea, www.qiagen.com) respectively, and single-strand cDNA was synthesized with random oligonucleotides using a reverse transcription system (Bioneer, www.bioneer.co.kr, E-3122 (M-MLV Reverse Transcriptase) based on a previously reported method [18]. Primers used in the experiment are listed in Appendix A. qRT-PCR analysis was performed according to the manufacturer’s instructions using LightCyclerR 480 Real Time PCR system (Roche, CA, United States). “Relative expression” levels were calculated using the 2^−ΔΔCT^ method, and *OsActin* (Q10DV7) was used as a reference gene in rice. All experiments included three biological replicates, each with two technical replicates.

### 2.8. Statistical Analysis 

Data collected in this study were analyzed by one-way analysis of variance (ANOVA) using the Statistical Analysis System (SAS version 9.4). Values are mean ± SE (*n* = 3) and statistical significance was set to *p* < 0.05 according to Duncan’s multiple range test [32]. 

## 3. Results

### 3.1. Expression Profile of OsOr Gene

The full-length cDNA of *OsOr* was cloned into PMD18-T vector after PCR analysis using gene specific primer sets, and confirmed by sequencing. This gene consists of a 996 bp ORF encoding a putative protein of 332 amino acids, and GenBank’s accession number is AK099767 (Appendix A). The OsOr protein had an estimated molecular mass of 34.3 kDa and pI of 8.36 (Appendix A). The genetic similarity between the sequences of *Os**Or* and *Or* genes of various plant species showed the highest homology with the *Or* gene of morning glory (*Ipomoea nil*) (TA6874_35883) and was a 97% identity at the amino acid level (Appendix A). In addition, *Os**Or* showed 73 - 80% sequence homology between several plant-derived *Or* genes, including the putative *Or* genes of tomato (*L. esculentum*), grape (*Vitis vinifera*), *Arabidopsis thaliana* (At5g61670), and cauliflower (*B. oleracea* var. *botrytis*) (Appendix A). The domain annotation of the OsOr protein contained a motif with two transmembrane domains, a plastid-targeting transit sequence, and a repeating cysteines (CxxCxGxGx) characteristic of the DnaJ protein known as a chaperone. To determine the expression pattern of *OsOr* in rice, we analyzed the expression profile of *OsOr* gene in roots, stems, leaves, panicles, and immature seeds using qRT-PCR. The results showed that *OsOr* gene was expressed in all tissues tested, and this gene expression was highly detected in stems and lowly detected in immature seeds (Figure 1A). In addition, we investigated the expression pattern of the *OsOr* gene in response to H_2_O_2_, PEG, heat, cold, ABA, and GA_3_ stresses. The results suggested that H_2_O_2_, PEG, heat, and cold stresses except GA_3_ stress induced the expression of *OsOr* at all treatment time points (Figure 1B). Under H_2_O_2_, PEG, and cold stress conditions, the expression level of *OsOr* was rapidly induced from the early stage and showed the plateau expressions for 3–24 h following the treatment. Under heat stress condition, the expression level of *OsOr* was rapidly induced, and gradually increased to 24 h of treatment. In addition, the cold stress condition slightly significantly induced the expression of *OsOr*, but the GA_3_ stress conditions did not show significant difference in the *OsOr* expression level. Therefore, the results of expression profile for *OsOr* indicated that *OsOr* gene might be important for plant tolerance against H_2_O_2_, PEG, heat, and cold stresses.

### 3.2. Phenotypic Analysis of Transgenic Rice 

To further study the function of the *OsOr* gene, we constructed *OsOr*-WT and *OsOr*-R115H under the control of the CaMV 35S promoter and introduced them in rice embryogenic callus using an *Agrobacterium*-mediated method (Figure 2A). To select transgenic plants with a single copy gene, ten T0 transformants with *OsOr*-WT and *OsOr*-R115H were tested through PCR analysis on whether T-DNA was inserted as well as a TaqMan PCR analysis. Four independent T3 homozygous transgenic lines, #7 and #9 for *OsOr*-WT, and #6 and #10 for *OsOr*-R115H, were selected and used for subsequent analysis (Appendix A; Figure 2B).

The expression level of *OsOr* gene was found to be statistically significantly higher in transgenic lines (#7, #9, #6, #10) than in NT plants (Figure 2C). In addition, the flower structure and pollen germination of transgenic plants were shown similar to those of NT plants (Figure 3A). In addition, the transgenic lines and NT plant showed similar responses to disease in the inoculation experiment with *Xanthomonas oryzae* pv. *oryzae (Xoo*) strain (Figure 3B). In order to investigate the morphological characteristics of rice grain in seed cross-sections, the results of scanning electron microscope (Quanta FEG 450, FEI Electron Microscopy) analysis showed that the seeds of the transgenic lines were almost similar to those of the NT plants (Figure 3C). As a result of examining starch content and amylose content of transgenic lines, there were no significant differences from NT plants (Figure 3D). In addition, statistical analysis showed that no significant differences were detected in transgenic lines (#7, #9, #6, #10) compared to NT plants, including shoot length, root length, seed setting rate, pollen fertility, 1000-grain weight, and yield per plant (Figure 4). Taken together, our results showed that overexpression of *OsOr*-WT and *OsOr*-R115H was not significant phenotypic alterations compared to NT plants. In addition, the carotenoid content between the transgenic lines and NT plants was analyzed using high performance liquid chromatography (HPLC) analysis. The results were almost similar in the investigated lines (Appendix A), implying that the *OsOr* gene was not directly affect to the carotenoid contents in rice [33].

### 3.3. Overexpression of OsOr-R115H Enhances Heat Stress Tolerance 

To examine whether *OsOr* plays a role in heat stress tolerance, we investigated heat tolerance of the transgenic lines (*OsOr*-WT (#7, #9), and *OsOr*-R115H (#6, #10)) together with WT plants. As shown in Figure 5A, no significant morphological differences of leaf discs were observed between transgenic and NT plants before heat stress treatment. However, after incubation at 48 °C for 24 h, NT plants showed severe membrane damages, as evident from the dark-brown coloration of leaf discs, whereas *OsOr*-WT and *OsOr*-R115H leaf discs displayed less membrane damage. *OsOr*-R96H transgenic plants showed greater heat stress tolerance than NT and *OsOr*-WT plants (Figure 5A). Our data showed that the oxidized DAB contents of NT leaves at 12 h after heat stress was 21.3 mg/g DW, significantly higher than those of transgenic lines (the oxidized DAB contents of *OsOr*-WT #7, #9, *OsOr*-R115H #6 and *OsOr*-R115H #10 lines were 10.9, 9.2, 8.7, and 7.7, respectively) (Figure 5B). In addition, we determined the proline and chlorophyll contents in the leaves from transgenic lines and NT plants. Our results showed no difference in proline and chlorophyll contents between NT plants and transgenic lines grown under normal conditions. On the other hand, the proline and chlorophyll contents of the transgenic lines (*OsOr*-WT and *OsOr*-R115H) after heat stress were significantly higher than NT plants (Figure 6A,B). Recently, it has reported that MDA and electrolyte leakage were strongly correlated with the degree of cell membrane damage under abiotic stress [34]. Thus, we measured the MDA and REL content from transgenic and wild type plants. The data displayed no obvious difference between transgenic lines and NT plants grown under normal conditions. However, the MDA and REL contents of *OsOr*-WT and *OsOr*-R115H transgenic plants were significantly lower than NT plants grown under heat stress conditions (Figure 6C,D). As a result of analyzing carotenoids after heat stress in the transgenic plants with *OsOr*-WT and *OsOr*-R115H, the carotenoid content was slightly decreased compared to the results under normal conditions after 24 h of heat stress treatment. However, there was no difference between NT plants and overexpression plants. These results suggest that the carotenoid accumulation pathway in rice is due to differences from that of sweet potatoes.

### 3.4. Transcription Analysis of Stress-Related Genes

To illustrate the regulation mechanism underlying the enhanced heat stress tolerance, we measured the expression of four genes encoding ROS scavenging enzymes (*OsCATA*, *OsCATB*, *OsAPX2*, *OsSOD-Cu/Zn*) and five genes (*OsLEA3*, *OsDREB2A*, *OsDREB1A*, *OsP5CS*, *SNAC1*) related to abiotic stress tolerance under normal growth and heat stress conditions using qRT-PCR (Figure 7 and Figure 8). There was no significant difference in the expression of all genes investigated under normal growth conditions. By contrast, the expression of all genes was remarkably higher in *OsOr*-WT and *OsOr*-R115H plants than in NT plants after heat stress. These results show that *OsOr* overexpression induces proteins involved in ROS scavenging pathways and defense mechanisms, thereby reducing oxidative damage and improving heat resistance.

## 4. Discussion

Plant orange (Or) proteins contain an N-terminal unknown region, transmembrane domains, and a C-terminal DnaJ-like domain, and meaningful progress has been made in understanding the biological function of the orange genes [8]. Some orange (*Or*) genes such as *BoOr* (ABH07405), *IbOr* (KX792094), *CmOr* (A0A0D3MU35), *AtOr* (Q8VYD8), *SIOr* (NP_001315338), and *DcOr* (KZN00418) showed resistance to various environmental stresses as well as carotenoid accumulation [8,9,10,11,12,13,14,15,16]. However, the clear biological functions of these genes are still unclear and require further studies. A single nucleotide substitution in the gene encoding CmOr is correlated with the orange color fruit. This substitution results in an Arg to His change at the 108th amino acid of CmOr [9]. In previous report, overexpression of AtOr^His^ (R90H), SbOr^His^ (R104H), and IbOr^His^ (R96H) with mutations at the site corresponding to the mutation in CmOr (golden SNP altering Arg to His) resulted in high total carotenoid levels and β-carotene accumulation [13,22,35]. In a sequence alignment, we found that OsOr possessed Arg at the 115th amino acid, corresponding to the 108th position of CmOr, 90th position of AtOR and 96th position of IbOr. Thus, to examine whether the substitution of the conserved Arg to His in OsOr promoted carotenoid accumulation, site-directed mutagenesis of *OsOr*-WT was performed to generate *OsOr*-R115H. In this study, we report the isolation, characterization, and physiological function of the orange (*Or*) gene from rice. Our results show that in the overexpressing plants, *OsOr*-WT and *OsOr*-R115H, carotenoid content was not changed compared to NT plants, but heat stress tolerance was enhanced. In addition, we did not find significant differences between *OsOr*-WT and *OsOr*-R115H. The reason why *OsOr*-R115H did not have a significant effect on carotenoids, unlike sweet potatoes, is thought to be due to a different mechanism of carotenoid accumulation, which is consistent with that previously reported by Yu el al. [33]. Evidence has shown that some *Or* genes are induced or inhibited expression by abiotic stress, and overexpression of these genes can increase the tolerance of transgenic plants to heat [8]. Similarly, our results showed rapid changes of *OsOr* gene expression in heat and PEG stress (Figure 1), suggesting *OsOr* might play a critical role in heat and PEG stress. In addition, overexpression of the golden SNP-carrying mutant *IbOr* allele was resulted to have enhanced abiotic stress tolerance as well as carotenoids accumulation [8]. In addition, Yu et al. (2021) reported that salt and cold stress resistance was reduced in Nipponbare rice plants overexpressing the *OsOr* gene [33]. Therefore, we constructed *OsOr* gene (*OsOr*-WT and *OsOr*-R115H) for overexpressing transgenic plants and tested their heat stress tolerance. Our data suggested that the overexpression of *OsOr*-WT and *OsOr*-R115H enhanced heat stress tolerance, and transgenic plants showed less membrane damage and aging compared to NT plants in Dongjin rice (Figure 5 and Figure 6). In general, when plants encountered abiotic stresses such as heat and salt, REL, proline, and MDA can react quickly to survive in extreme environmental conditions [34,36]. Therefore, the REL, an important parameter reflecting the degree of cell membrane damage, was significantly lower in the leaves of the transgenic plants than in the leaves of NT plants (Figure 6D). These results suggested that the *OsOr* gene might protect cell membrane integrity of plants in response to heat stress. Mellacheruvu et al. [37] have reported that abiotic stress on plants causes lipid peroxidation, resulting in MDA accumulation. Our data displayed that the MDA content was higher in NT plants than in transgenic plants with *OsOr-WT* and *OsOr-R115H* (Figure 6C), which shows that heat stress has more damage to NT plants compared to transgenic plants. Furthermore, proline has been reported to act as a stabilizer to protect plants from abiotic stress damage [38]. In our data, the proline of the transgenic plant with *OsOr*-WT and *OsOr*-R115H accumulated more compared with NT plants under heat stress conditions. These results suggested that proline might be responsible for enhancing heat stress tolerance shown by the transgenic plants overexpression *OsOr*. These results showed that enhanced heat stress tolerance of the transgenic plants with *OsOr*-WT and *OsOr*-R115H was at least partially related to reduced MDA content, reduced REL, and increased proline content. In addition, biochemical factors such as MDA, REL, etc. mentioned above have also contributed to significantly inducing expression of abiotic stress-response genes [39,40]. In our study, the expression level of four genes related to ROS-scavenging enzymes (*OsCATA*, *OsCATB*, *OsAPX2* and *OsSOD-Cu/Zn*) and six genes in response to abiotic stress (*OsLEA3*, *OsDREB2A*, *OsDREB1A*, *OsP5CS*, *SNAC1*) were more highly expressed in the transgenic lines (*OsOr*-WT #7, *OsOr*-WT #9, *OsOr*-R115H #6 and *OsOr*-R115H #10) than in NT plants under heat stress conditions (Figure 7 and Figure 8). Sirko et al. [41] reported that the function of *OsCATA* gene plays a role in protecting plants from damage caused by reactive oxygen species (ROS). It has been demonstrated that there is a positive correlation between the expression of *OsP5CS* gene and the accumulation of proline, and overexpression of *OsP5CS* increases resistance to abiotic stress in transgenic plants [42]. In addition, *OsLEA3*, *OsDREB2A*, *OsDREB1A*, and *SNAC1* genes were reported to be sensitive to abiotic stress [43,44,45,46,47]. It was reported that the plant hormone ABA is produced by five 9-cis-epoxycarotenoid dioxygenases (NCEDs), and strigolactones are synthesized by the enzymes CCD7 and CCD8 [48]. However, because of examining the expression of *OsOr* gene, it was slightly upregulated by ABA stress, and there was no difference in carotenoid content in transgenic plants. Taken together, the results showed that the transgenic lines further enhanced heat stress tolerance, perhaps by strengthening the expression levels of genes related to ROS-scavenging enzymes and responding to abiotic stress under heat stress conditions. The current study also demonstrated that the *OsOr* gene could serve as a candidate for breeding heat stress tolerant varieties in rice. Future research should further explore *OsOr* gene regulatory network and functionality using CRISPR/Cas9 system mediated base-editing technique, etc., in greater depth to unravel the mechanism for controlling heat stress tolerance in rice.

## Figures and Tables

**Figure 1 genes-12-01891-f001:**
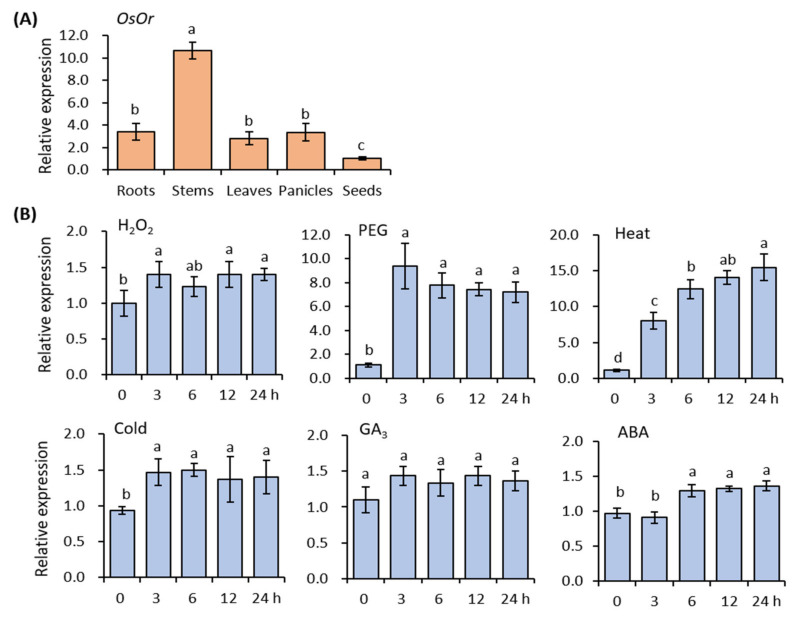
Expression analysis of the *OsOr* gene in T_3_ generation. (**A**) qRT-PCR analysis of *OsOr* gene expression in various tissues. Bars show means ± SD with three biological replicates. (**B**) *OsOr* expression analysis (qRT-PCR) in leaves of 2-week-old rice seedlings subjected to H_2_O_2_, 20% PEG 6000, heat, cold, ABA and GA_3_ treatments, respectively. Bars show standard deviations of the replicates. Data are means ± SE (*n* = 3). Different letters above the columns indicate significant differences between lines (*p* < 0.05).

**Figure 2 genes-12-01891-f002:**
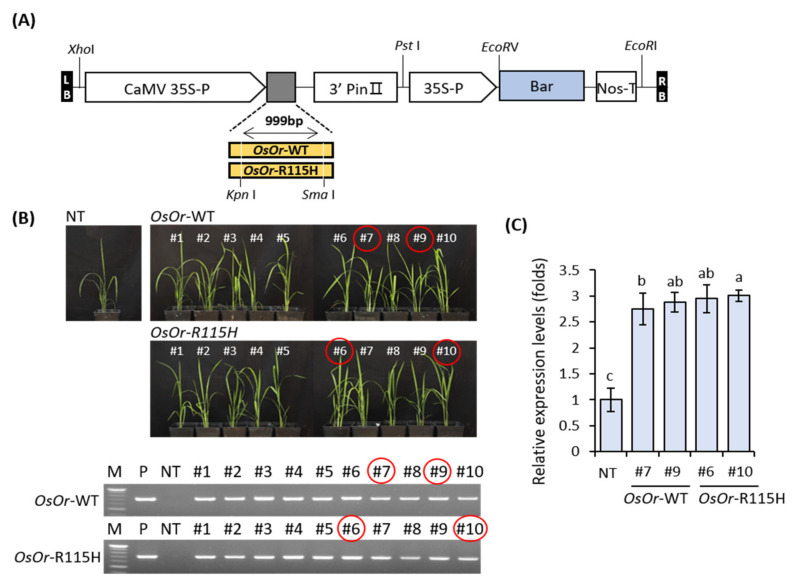
Generation of transgenic rice plants overexpressing *OsOr*-WT and *OsOr*-R115H. (**A**) Schematic representation of the constructs used for the production of transgenic rice plants overexpressing *OsOr*-WT and *OsOr*-R115H. (**B**) Photographs of aerial plant parts of 1-month-old NT, *OsOr*-WT (#7, #9) and *OsOr*-R115H (#6, #10) plant lines and genotyping transgenic plants using bar-specific primers. P, positive control. (**C**) Transcript levels of *OsOr* in transgenic rice plants. Data are means ± SE (n = 3). Different letters above the columns indicate significant differences between lines (*p* < 0.05).

**Figure 3 genes-12-01891-f003:**
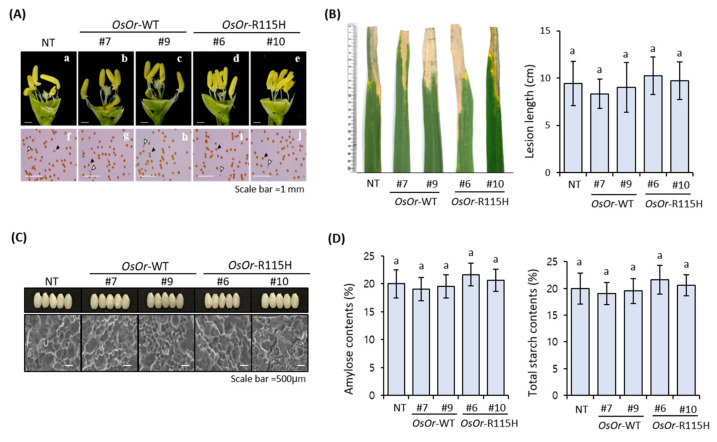
Phenotype of NT plants and transgenic rice lines overexpressing *OsOr* subjected to normal conditions. (**A**) Figure 7. (#9) and *OsOr*-R115H (#6, #10) lines. (**B**) Lesions and lesion length of NT plants, *OsOr*-WT (#7, #9) and *OsOr*-R115H (#6, #10) lines inoculated with *Xanthomonas oryzae* pv. *oryzae* (*Xoo*) strain for 2 weeks. (**C**) Seeds morphology and SEM analysis and (**D**) Amylose contents (%) and total starch contents (%) in T_3_ generation. Data are means ± SE (*n* = 3). Different letters above the columns indicate significant differences between lines (*p* < 0.05).

**Figure 4 genes-12-01891-f004:**
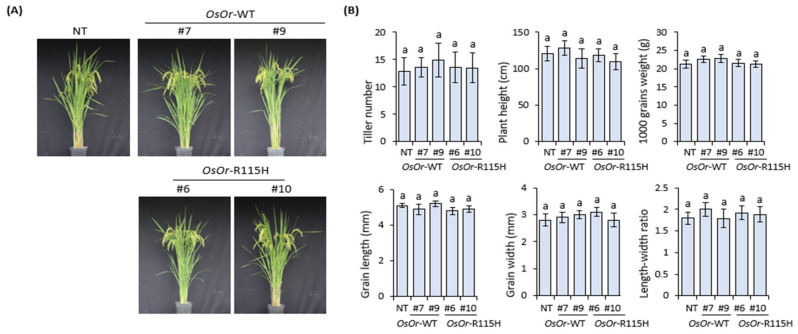
Agronomic traits of NT plants and transgenic rice lines overexpressing *OsOr* subjected to normal conditions. Morphology of mature plants (**A**), Tiller numbers, Plant height, 1000 grains weight, Grain length, Grain width and length width rate (**B**). Different letters between *OsOr*-WT, *OsOr*-R115H, and NT plants indicate significant differences (*p* < 0.05) according to Duncan test (mean ± SE, *n* = 3).

**Figure 5 genes-12-01891-f005:**
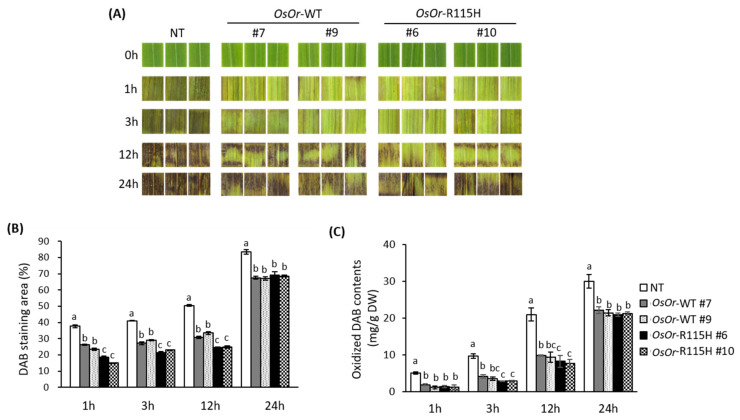
Analysis of heat stress tolerance in transgenic rice plants using leaf discs. (**A**) Images of leaf discs and quantification of ROS production in leaves using DAB staining. (**B**) DAB stained areas were measured and were quantified using ImageJ software in leaves at 1 h, 3 h, 12 h after 48 °C heat treatment. (**C**) H_2_O_2_ production after 1 h, 3 h and 12 h of treatment with heat stress. NT, *OsOr*-WT and *OsOr*-R115H transgenic rice cell lines were treated with a DAB-HCl solution and oxidized DAB content was compared. Data are means ± SE (*n* = 3). Different letters above the columns indicate significant differences between lines (*p* < 0.05).

**Figure 6 genes-12-01891-f006:**
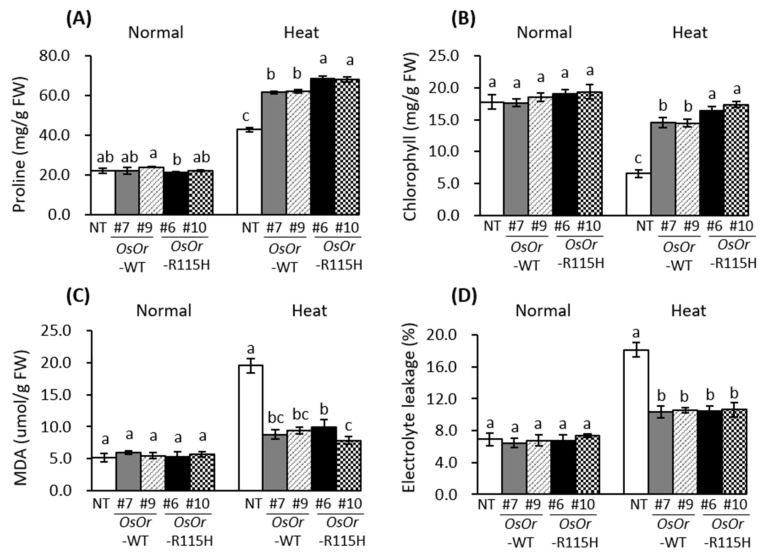
Contents of (**A**) Proline (mg/g FW), (**B**) chlorophyll (mg/g FW), (**C**) lipid peroxidation (malondialdehyde (MDA), μmol/g FW) level and (**D**) electrolyte leakage of the NT plants and transgenic rice lines overexpressing *OsOr* subjected to normal and heat stress conditions. Data are means ± SE (*n* = 3). Different letters above the columns indicate significant differences between lines (*p* < 0.05).

**Figure 7 genes-12-01891-f007:**
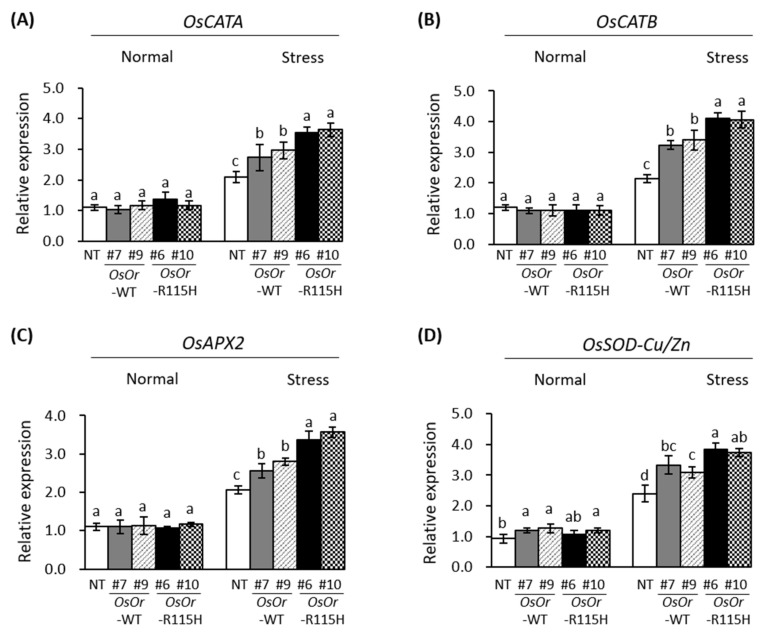
Expression levels of (**A**) *OsCATA*, (**B**) *OsCATB*, (**C**) *OsAPX2*, and (**D**) *OsSOD*-*Cu/Zn* genes in the NT plants and transgenic rice lines overexpressing *OsOr* subjected to normal and heat stress conditions. Data are means ± SE (*n* = 3). Different letters above the columns indicate significant differences between lines (*p* < 0.05).

**Figure 8 genes-12-01891-f008:**
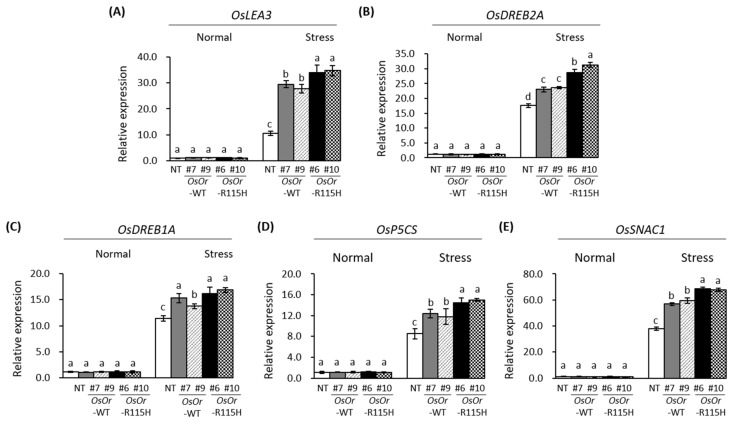
Expression levels of (**A**) *OsLEA3*, (**B**) *OsDREB2A*, (**C**) *OsDREB1A*, (**D**) *OsP5CS* and (**E**) *OsSNAC1* genes in the NT and transgenic rice lines overexpressing *OsOr* under normal and heat stress conditions. Data are means ± SE (*n* = 3). Different letters above the columns indicate significant differences between lines (*p* < 0.05).

## Data Availability

Not applicable.

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
