# Peer review of "Overexpression of Orange Gene (OsOr-R115H) Enhances Heat Tolerance and Defense-Related Gene Expression in Rice (Oryza sativa L.)"

_genes, 2021, doi:10.3390/genes12121891_

Round 1

Reviewer 1 Report

The study characterizes the expression of the orange gene (Or) in rice, and its relationship to heat stress. Their study provides potential application of OsOr gene in heat resistant cultivar breeding. However, this article needs to be revised with major revisions. The article needs extensive editing to fix numerous grammatical and formatting errors.

My comments (not including all the grammar errors):

Page 2 line 68: Need to add scientific name of sweetpotato.

Page 2 line 72 and line 77: The abbreviations such as “ORF”, “MAD”, and “REL should be spelled out when they first appear.

Page 2 line 84: At what stages are these tissues sampled?

Page 2 line 84-86: This sentence is confusing

Page 2 line 91: “were amplified” should be “was amplified”

Page 3 line 101: Use “PCR” instead of “genomic PCR”

Page 3 line 111: Use scientific name for “Xoo

Page 3 line 118: Re-write this sentence.

Page 3 line 121: Delete “heat-treated”
Page 3 line 126-127: Please briefly describe the purpose of this analysis and the methodology.

Page 3 line 129: Use “an RNeasy” instead of “a RNeasy”

Page 3 line 135: Missing “.” Before the “Relative expression”

Page 4 line 147: It’s necessary to conduct phylogenetic analysis and protein domain annotation of Or gene.

Page 4 line 156-160: The description is contradictory to the Figure 1B. Apparently, the gene showed a continuous rise under heat stress conditions, not under PEG stress conditions. Besides, cold treatment also significantly induced the expression of OsOr, but the authors ignored it in result and discussion.

Page 5: Missing figure caption for Figure 3D.

Page 6 line 193: Use “were” instead of “was”

Page 6 line 194 - 196: Too many grammar errors and hard to read. Re-write this sentence.

Page 6 line 221 and 222: Use “oxidized” instead of “oxidazed”. Same for the Figure 5.

Page 7 line 248: Be specific, what genes showed no significant differences and what genes were significantly different.

Page 8: The Figure 6 showed results collected from 4 time points. But in the Methods part, they only mentioned “heat stress treatment (48 °C) for 24 h.”

Page 8 and 9: Figure 7 and Figure 8. Add the gene name to each plot. The different bar patterns provide no information here, it’s better to integrate the expression of different genes to 1 barplot.

Page 9 line 279: The author didn’t include carotenoids analysis result in this manuscript.  Should add the analysis of carotenoid levels or carotenoid genes expression. The studies by Bai et al., 2014, “An in vitro system for the rapid functional characterization of genes involved in carotenoid biosynthesis and accumulation” and Bai et al., 2016, “Bottlenecks in carotenoid biosynthesis and accumulation in rice endosperm are influenced by the precursor–product balance” showed that the overexpression of AtOr-His in rice increased the carotenoid accumulation. It will be interesting to analyze and discuss the difference. But these results are not mentioned in this paper.

Page 9 line 288: There is not significant difference between OsOr-WT and OsOr-R115H overexpression lines. The author should discuss it in the discussion. Also, they should describe how and why they construct the OsOr-R115H overexpression lines in the Materials and Methods.

Page 9 line 293: Allow ...? to survive

Page 10 line 296: Use “suggested” instead of “suggesting”

Page 10 line 301 and 302: Use “has” instead of “have”

Supplementary Figure S1: missing space between “Or gene” and “(Os02g0651300)”

Author Response

Response to Comments

We appreciate the comments that the reviewers have given in our manuscript and the constructive criticism they have given. We have carefully reviewed the comments and have revised the manuscript accordingly. We believe that these changes have clearly improved our manuscript.  

Reviewer 1

Page 2 line 68: Need to add scientific name of sweetpotato.

--- Thank you for the critical comments. Authors have added sweetpotato (Ipomoea batatas [L.] Lam).

Page 2 line 72 and line 77: The abbreviations such as “ORF”, “MAD”, and “REL should be spelled out when they first appear.

--- Thank you for the critical review. We have revised it according to the comments.

Page 2 line 84: At what stages are these tissues sampled?

--- We sampled from plants at 7 days after pollination. We have described these in the Line 85~89 as follows: The expression levels of OsOr was investigated in roots, leaves, stems, panicles and immature seeds at 7 days after pollination in Dongjin cultivar.

Page 2 line 84-86: This sentence is confusing

--- Thank you for the critical review. Authors have revised according to the comments as follows: The expression levels of OsOr was investigated in roots, leaves, stems, panicles and immature seeds at 7 days after pollination in Dongjin cultivar. The expression pattern of the OsOr in response to abiotic stresses including 48 °C heat, 200 mM NaCl and 20 % PEG6000 drought 100 μM GA3, 100 μM ABA and 4 °C cold was monitored as previously reported by Park et al. [17].

Page 2 line 91: “were amplified” should be “was amplified”

--- We have revised it as follows: “was amplified” at Line 94.

Page 3 line 101: Use “PCR” instead of “genomic PCR”

--- Thank you for the critical review. Authors have revised according to the comments.

Page 3 line 111: Use scientific name for “Xoo

--- Thank you for the critical review. Authors have added scientific name according to the comments as follows: Xanthomonas oryzae pv. oryzae (Xoo)

Page 3 line 118: Re-write this sentence.

--- Thank you for the critical review. Authors have re-written the sentence as follows:

To induce heat stress, leaf discs were excised from the 4th leaves of 2-week-old transgenic and NT rice plants, and treated to high temperature (48 °C) in the dark for 24 h.

Page 3 line 121: Delete “heat-treated”

--- Thank you for the critical review. Authors have deleted it according to the comments.

Page 3 line 126-127: Please briefly describe the purpose of this analysis and the methodology.

--- Thank you for the critical review. References were included, and the authors measured them according to the methods described in the references.

Page 3 line 129: Use “an RNeasy” instead of “a RNeasy”

--- Thank you for the critical review. Authors have revised according to the comments.

Page 3 line 135: Missing “.” Before the “Relative expression”

--- Thank you for the critical review. Authors have revised according to the comments.

Page 4 line 147: It’s necessary to conduct phylogenetic analysis and protein domain annotation of Or gene.

---- Thank you for the critical review. We have revised it at Lines 159~169.

Page 4 line 156-160: The description is contradictory to the Figure 1B. Apparently, the gene showed a continuous rise under heat stress conditions, not under PEG stress conditions. Besides, cold treatment also significantly induced the expression of OsOr, but the authors ignored it in result and discussion.

--- Thank you for the critical review. We have revised according to the comments as described in Lines 175~184:

The results suggested that H2O2, PEG, heat, and cold stresses except GA3 stress induced the expression of OsOr at all treatment time points (Figure 1B). Under H2O2, PEG, and cold stress conditions, the expression level of OsOr was rapidly induced from the early stage and showed the plateau expressions for 3-24 h following the treatment. Under heat stress condition, the expression level of OsOr was rapidly induced, and gradually increased to 24 h of treatment. In addition, the cold stress condition slightly significantly induced the expression of OsOr, but the GA3 stress conditions did not show significant difference in the OsOr expression level. Therefore, the results of expression profile for OsOr indicated that OsOr gene might be important for plant tolerance against H2O2, PEG, heat, and cold stresses.

Page 5: Missing figure caption for Figure 3D.

--- Thank you for the critical review. Authors have added figure caption in Figure 3D.

Page 6 line 193: Use “were” instead of “was”

--- Thank you for the critical review. Authors have revised according to the comments.

Page 6 line 194 - 196: Too many grammar errors and hard to read. Re-write this sentence.

--- Thank you for the critical review. Authors have re-write according to the comments as follows:

In addition, the transgenic lines and NT plant showed similar responses to disease in the inoculation experiment using Xanthomonas oryzae pv. oryzae (Xoo) strain

Page 6 line 221 and 222: Use “oxidized” instead of “oxidazed”. Same for the Figure 5.

--- Thank you for the critical review. Authors have revised according to the comments.

Page 7 line 248: Be specific, what genes showed no significant differences and what genes were significantly different.

--- Thank you for the critical review. Because there was a difference in expression among the selected genes, we did not explain each gene.

Page 8: The Figure 6 showed results collected from 4 time points. But in the Methods part, they only mentioned “heat stress treatment (48 °C) for 24 h.”

--- Thank you for the critical review. Expression analysis of the OsOr gene of the transformant was limited to heat stress treatment.

Page 8 and 9: Figure 7 and Figure 8. Add the gene name to each plot. The different bar patterns provide no information here, it’s better to integrate the expression of different genes to 1 barplot.

--- Thank you for the critical review. We have added gene names into figure 7 and figure 8.

Page 9 line 279: The author didn’t include carotenoids analysis result in this manuscript.  Should add the analysis of carotenoid levels or carotenoid genes expression. The studies by Bai et al., 2014, “An in vitro system for the rapid functional characterization of genes involved in carotenoid biosynthesis and accumulation” and Bai et al., 2016, “Bottlenecks in carotenoid biosynthesis and accumulation in rice endosperm are influenced by the precursor–product balance” showed that the overexpression of AtOr-His in rice increased the carotenoid accumulation. It will be interesting to analyze and discuss the difference. But these results are not mentioned in this paper.

--- Thank you for the critical review. Authors have changed according to the comments

Our results show that in plants overexpressing OsOr-WT and OsOr-R115H, carotenoid content was not changed compared to NT plants, but heat stress tolerance was enhanced. Also, we did not find significant differences between OsOr-WT and OsOr-R115H. The reason why OsOr-R115H did not have a significant effect on carotenoids, unlike sweet potatoes, is thought to be due to a different carotenoid accumulation mechanism.

Page 9 line 288: There is not significant difference between OsOr-WT and OsOr-R115H overexpression lines. The author should discuss it in the discussion. Also, they should describe how and why they construct the OsOr-R115H overexpression lines in the Materials and Methods.

--- Thank you for the critical review. Authors have revised according to the comments.

Page 9 line 293: Allow ...? to survive

--- Thank you for the critical review. Authors have revised it at Line 319.

Page 10 line 296: Use “suggested” instead of “suggesting”

--- Thank you for the critical review. Authors have revised according to the comments.

Page 10 line 301 and 302: Use “has” instead of “have”

--- Thank you for the critical review. Authors have revised it at Line 327.

Supplementary Figure S1: missing space between “Or gene” and “(Os02g0651300)”

--- Thank you for the critical review. Authors have revised according to the comments.

Reviewer 2 Report

In this manuscript, Jung et al over-expressed the rice OsOr gene and found better heat stress tolerance after a short time heat stress. Although the data provide new insight into the OsOr gene, I found the manuscript lack key experiments and does not go deep enough into the mechanism. I listed my major and minor concerns below.

Major points:

  1. On line 72, 73, the reasoning to do the OsOr-R115H should be clearer. Does the author want to test whether the R115H will increase the carotenoid contents in rice? If so, what is the result on the carotenoid content? Does OsOr gene act the same as the sweet potato ortholog?
  2. I didn’t find too much difference between OsOr-WT and OsOr-R115H based on the results. Is this expected? If OsOr-R115H does not have a big effect on carotenoid, unlike the sweet potato, why? Please discuss.
  3. On line 160, there is no effect of ABA which is a key hormone in stress response, why? Please discuss.
  4. For Figure 5B, I guess the unit is per gram of fresh weight. Why use fresh weight? As the heat stress goes on, the plants will lose water. Is it possible that the observation of the DAB increase/not increase, was just because of water loss, but not cell damage? Is it better to use the dry weight? Similar issues go with Figure 6 which uses fresh weight.
  5. To better support the result of section 3.3, the authors should measure the agronomic traits for OsOr-WT and OsOR-R115, like figure 4, after the heat stress. Otherwise, the conclusion is just based on very short-term treatment and minimum practical usage.
  6. Figure 8 showed the stress marker genes are highly induced in OsOr transgenic plants. The DREBs are also markers for drought and ABA, but based on figure 1, the authors did not observe a significant effect of ABA treatment on OsOr. Does it mean the OsOr does not regulate by ABA? Based on figure 1B, the H2O2 effect is mild too. Please discuss what signal under PEG (which is often used to mimic drought) and Heat that OsOr senses.
  7. I suggest the authors do an RNA-Seq experiment OE-OsOr vs NT to go deeper into the mechanism of OsOr on the short-term heat stress tolerance. If the authors are going to do (or already did) the field long-term experiment (measuring grain yield, biomass etc), which I strongly suggest, the samples can be collected from the field too.

Minor points:

  1. On line 63, the TAIR ID for AtOr should be written out, although it’s at the end of the manuscript.
  2. The reason to do the DAB experiment should be clearly explained. The method should be briefly described.
  3. On line 256, the methods for REL and MDA content measurement should be described.
  4. On line 131, the catalog number for the Bioneer system should be clearly written out.
  5. On line 136, the OsActin gene ID should be clearly written out.
  6. On line 140, the SAS version, the function code should be clearly written.
  7. I think Section “3.2. Phenotypic analysis of transgenic rice” should be greatly reduced and Figure 3 should be supplementary. I appreciate the authors measured the phenotypes and found the transgenic lines do not change phenotypes. However, I don’t think it’s necessary to include a whole result section and the main figure for this.
  8. On line 215, need to explain the experiment used for Figure 5A. It seems the author's conclusion is based on eye observation. Is it possible to use ImageJ or Python to quantify the “brown” pixel and do statistical tests?
  9. On line 221, it is not clear why the authors did the DAB experiment. Also, the unit for the numbers should be include. I think it should be 21.3 mg/g fw?
  10. In line 331, the method for MSA is not in the method section.
  11. The gene IDs should be listed as a column for Table S1, for example, Os02g0651300. This is related to my minor point 5.
  12. On line 140, the author wrote the statistical test is “Duncan multiple range test”. But duncan multiple range test is just the post-hoc test. Did the author do ANOVA first? If so, what function? The parameter used should also be included. This is related to my minor point 6.

Author Response

Response to Comments

We appreciate the comments that the reviewers have given in our manuscript and the constructive criticism they have given. We have carefully reviewed the comments and have revised the manuscript accordingly. We believe that these changes have clearly improved our manuscript.  

Reviewer 2

Comments and Suggestions for Authors

In this manuscript, Jung et al over-expressed the rice OsOr gene and found better heat stress tolerance after a short time heat stress. Although the data provide new insight into the OsOr gene, I found the manuscript lack key experiments and does not go deep enough into the mechanism. I listed my major and minor concerns below.

Major points:

  1. On line 72, 73, the reasoning to do the OsOr-R115H should be clearer. Does the author want to test whether the R115H will increase the carotenoid contents in rice? If so, what is the result on the carotenoid content? Does OsOr gene act the same as the sweet potato ortholog?

--- Thank you for the critical review. Authors have revised the parts as follows:

(1) Lines 72~78:

In this study, to understand the physiological role of the Or gene in rice, the carotenoid content and enhances abiotic stress tolerance in plants overexpressing OsOr-WT and OsOr-R115H were investigated. Our results show that in plants overexpressing OsOr-WT and OsOr-R115H, carotenoid content was not changed compared to NT plants, but heat stress tolerance was enhanced. In addition, these lines showed a high level of transcription of genes related to ROS scavenging enzymes under stress conditions.

(2) Lines 121~125:

2.4. Analysis of carotenoid contents

Carotenoids were extracted from rice leave tissues using 0.01% solution of butylatedhydroxytoluene in acetone. and analyzed using an Agilent 1100 HPLC system (HewlettePackard, Palo Alto, CA, USA) according to the method described by Lim et al. [28].

(3) Lines 228~231: Results,

In addition, the carotenoid content between the transgenic lines and NT plants was analyzed using high performance liquid chromatography (HPLC) analysis. The results were almost similar in the investigated lines (Supplementary Table 1), implying that the OsOr gene was not directly affect to the carotenoid contents in this experiment. [33]

  1. I didn’t find too much difference between OsOr-WT and OsOr-R115H based on the results. Is this expected? If OsOr-R115H does not have a big effect on carotenoid, unlike the sweet potato, why? Please discuss.

--- Thank you for the critical review. Authors have revised in Result and Discussion parts as follows:

(1) Result:

In addition, the carotenoid content between the transgenic lines and NT plants was analyzed using high performance liquid chromatography (HPLC) analysis. The results were almost similar in the investigated lines (Supplementary Table 1), implying that the OsOr gene was not directly affect to the carotenoid contents in this experiment. [33]

(2) Discussion

Our results show that in plants overexpressing OsOr-WT and OsOr-R115H, carotenoid content was not changed compared to NT plants, but heat stress tolerance was enhanced. Also, we did not find significant differences between OsOr-WT and OsOr-R115H. The reason why OsOr-R115H did not have a significant effect on carotenoids, unlike sweet potatoes, is thought to be due to a different carotenoid accumulation mechanism, implying that the OsOr gene was not directly affect to the carotenoid contents in this experiment. [33]

  1. On line 160, there is no effect of ABA which is a key hormone in stress response, why? Please discuss.

--- Thank you for the critical review. We have revised as follows: As the reviewer pointed out, we tried the ABA experiment again. In the ABA re-experiment data, except for those with large and small variations, the data was reconstructed and converted into data (Fig. 1B).

Sentences with ABA processing: Under ABA stress conditions, the expression of OsOr was downregulated at 3 h, and slightly up-regulated by ABA from 6 to 24 h

  1. For Figure 5B, I guess the unit is per gram of fresh weight. Why use fresh weight? As the heat stress goes on, the plants will lose water. Is it possible that the observation of the DAB increase/not increase, was just because of water loss, but not cell damage? Is it better to use the dry weight? Similar issues go with Figure 6 which uses fresh weight.

--- Thank you for the critical review. The reviewer mentioned Figure 5B, but Figure 5B is about the DAB staining area in percent (%). In regard to Figure 5C, it was measured on a dry sample, but an error was made in the display.

  1. To better support the result of section 3.3, the authors should measure the agronomic traits for OsOr-WT and OsOR-R115, like figure 4, after the heat stress. Otherwise, the conclusion is just based on very short-term treatment and minimum practical usage.

--- Thank you for the critical review. We understand the reviewer's point, but in our experiments, the agronomic traits could not be measured because the plants did not grow after heat treatment.

  1. Figure 8 showed the stress marker genes are highly induced in OsOr transgenic plants. The DREBs are also markers for drought and ABA, but based on figure 1, the authors did not observe a significant effect of ABA treatment on OsOr. Does it mean the OsOr does not regulate by ABA? Based on figure 1B, the H2O2 effect is mild too. Please discuss what signal under PEG (which is often used to mimic drought) and Heat that OsOr senses.

--- Thank you for the critical review. Sentences with ABA processing: Under ABA stress conditions, the expression of OsOr was downregulated at 3 h, and slightly up-regulated by ABA from 6 to 24 h

--- Authors have re-written the sentence as follows:

It was reported that the plant hormone ABA is produced by five 9-cis-epoxycarotenoid dioxygenases (NCEDs), and strigolactones are synthesized by the enzymes CCD7 and CCD8 [47]. However, as a result of examining the expression of OsOr gene, it was slightly upregulated by ABA stress, and there was no difference in carotenoid content in transgenic plants.

  1. I suggest the authors do an RNA-Seq experiment OE-OsOr vs NT to go deeper into the mechanism of OsOr on the short-term heat stress tolerance. If the authors are going to do (or already did) the field long-term experiment (measuring grain yield, biomass etc), which I strongly suggest, the samples can be collected from the field too.

--- We are grateful for the critical review. Our future plans are for RNA-seq as pointed out by the reviewer.

Minor points:

  1. On line 63, the TAIR ID for AtOr should be written out, although it’s at the end of the manuscript.

--- Thank you for the critical review. Authors have revised it at Line 64.

  1. The reason to do the DAB experiment should be clearly explained. The method should be briefly described.

--- Thank you for the critical review. Authors have added the purpose of the DAB experiment at Line 133 according to the comments.

  1. On line 256, the methods for REL and MDA content measurement should be described.

--- Thank you for the critical review. References were included, and the authors measured them according to the methods described in the references.

  1. On line 131, the catalog number for the Bioneer system should be clearly written out.

--- Thank you for the critical review. Authors have revised it according to the comments.

  1. On line 136, the OsActin gene ID should be clearly written out.

--- Thank you for the critical review. Authors have revised it according to the comments.

  1. On line 140, the SAS version, the function code should be clearly written.

--- Thank you for the critical review. Authors have written as follows: Statistical Analysis System (SAS version 9.4).

  1. I think Section “3.2. Phenotypic analysis of transgenic rice” should be greatly reduced and Figure 3 should be supplementary. I appreciate the authors measured the phenotypes and found the transgenic lines do not change phenotypes. However, I don’t think it’s necessary to include a whole result section and the main figure for this.

--- Thank you for the critical review. In transgenic breeding, the equivalence of the transformant with the control plant is very important. Therefore, the results are described in the text.

  1. On line 215, need to explain the experiment used for Figure 5A. It seems the author's conclusion is based on eye observation. Is it possible to use ImageJ or Python to quantify the “brown” pixel and do statistical tests?

--- Thank you for the critical review. Added data converted to Image J, and authors have re-written the sentence as follows:

--- However, after incubation at 48 °C for 24 h, NT plants showed severe membrane damages, as evident from the dark-brown coloration of leaf discs in ImageJ, whereas OsOr-WT and OsOr-R115H leaf discs displayed less membrane damage. OsOr-R96H transgenic plants showed greater heat stress tolerance than NT and OsOr-WT plants (Figure 5A).

  1. On line 221, it is not clear why the authors did the DAB experiment. Also, the unit for the numbers should be include. I think it should be 21.3 mg/g fw?

--- Thank you for the critical review. Authors have written according to the comments.

--- Our data showed that the oxidized DAB contents of NT leaves at 12 h after heat stress was 21.3 mg/g dw, significantly higher than those of transgenic lines (the oxidized DAB contents of OsOr-WT #7, #9, OsOr-R115H #9 and OsOr-R115H #10 lines were 10.9, 9.2, 8.7 and 7.7, respectively) (Figure 5B).

  1. In line 331, the method for MDA is not in the method section.

--- Thank you for the critical review. References were included, and the authors measured them according to the methods described in the references.

-- Briefly, 500mg of rice leaves were pulverize in 2 mL of the chilled reagent [0.25% (w/v) thiobarbituric acid in 10% (w/v) trichloroacetic acid]. The extracts were treated at 100°C for 30 min, cooled at room temperature and centrifuged at 12,000 g for 15 min. The absorbance of the supernatant was measured at 450, 532 and 600 nm. The MDA content was calculated according to the formula: 6.45× (OD532-OD600)-0.559×OD450 [29,30,31].

  1. The gene IDs should be listed as a column for Table S1, for example, Os02g0651300. This is related to my minor point 5.

--- Thank you for the critical review. Authors have added gene ID according to the comments.

  1. On line 140, the author wrote the statistical test is “Duncan multiple range test”. But duncan multiple range test is just the post-hoc test. Did the author do ANOVA first? If so, what function? The parameter used should also be included. This is related to my minor point 6.

--- Thank you for the critical review. Authors have written it at Line as follows: Data collected in this study were analyzed by one-way analysis of variance (ANOVA) using the Statistical Analysis System (SAS version 9.4).

Round 2

Reviewer 1 Report

The authors addressed all my comments. I have no further objections. However, the the writing may need fine tuning. For example, P4L184 has a grammar error: "The domain annotation of the OsOr protein was contained a motif ..." 

I wish the authors could go through the whole text once more before the publication.

Author Response

Responses to Reviewer’s comments:

We appreciate the comments that the reviewers have given in our manuscript and the helpful criticism the reviewer have given. We have carefully reviewed the comments and have revised the manuscript accordingly. We believe that these changes have clearly improved our manuscript.

Thank you for kind comments.

The authors addressed all my comments. I have no further objections. However, the the writing may need fine tuning. For example, P4L184 has a grammar error: "The domain annotation of the OsOr protein was contained a motif..”

I wish the authors could go through the whole text once more before the publication.

--- Thank you for the reviewer’s comments. We have revised it in the lines 179-182 as follows:

The domain annotation of the OsOr protein contained a motif with two transmembrane domains, a plastid-targeting transit sequence, and a repeating cysteines (CxxCxGxGx) characteristic of the DnaJ protein known as a chaperone.

--- Also, we have reviewed and revised all sentences in this manuscript.
